# Does Protein Glycation Impact on the Drought-Related Changes in Metabolism and Nutritional Properties of Mature Pea (*Pisum sativum* L.) Seeds?

**DOI:** 10.3390/ijms21020567

**Published:** 2020-01-15

**Authors:** Tatiana Leonova, Veronika Popova, Alexander Tsarev, Christian Henning, Kristina Antonova, Nadezhda Rogovskaya, Maria Vikhnina, Tim Baldensperger, Alena Soboleva, Ekaterina Dinastia, Mandy Dorn, Olga Shiroglasova, Tatiana Grishina, Gerd U. Balcke, Christian Ihling, Galina Smolikova, Sergei Medvedev, Vladimir A. Zhukov, Vladimir Babakov, Igor A. Tikhonovich, Marcus A. Glomb, Tatiana Bilova, Andrej Frolov

**Affiliations:** 1Department of Biochemistry, St. Petersburg State University, 199004 St. Petersburg, Russia; tanyaleonova2710@gmail.com (T.L.); veronika.chantseva@gmail.com (V.P.); alexandretsarev@gmail.com (A.T.); krezistina@gmail.com (K.A.); vikhnina@gmail.com (M.V.); oriselle@yandex.ru (A.S.); dinastiyakate@gmail.com (E.D.); tgrishina@mail.ru (T.G.); 2Department of Bioorganic Chemistry, Leibniz Institute of Plant Biochemistry, 06120 Halle, Germany; mandy.dorn@ipb-halle.de; 3Department of Plant Physiology and Biochemistry, St. Petersburg State University, 199034 St. Petersburg, Russia; shiroglazovaolga@gmail.com (O.S.); galina.smolikova@gmail.com (G.S.); sergei.medvedev.spb@gmail.com (S.M.); 4Institute of Chemistry – Food Chemistry, Martin-Luther University Halle-Wittenberg, 06120 Halle, Germany; christian.henning@chemie.uni-halle.de (C.H.); tim.baldensperger@chemie.uni-halle.de (T.B.); markus.glomb@chemie.uni-halle.de (M.A.G.); 5Research Institute of Hygiene, Occupational Pathology and Human Ecology, 188663 Leningrad Oblast, Russia; nadin-r@mail.ru (N.R.); vbabakov@gmail.com (V.B.); 6Postovsky Institute of Organic Synthesis of Ural Division of Russian Academy of Sciences, 620137 Yekaterinburg, Russia; 7Department of Metabolic and Cell Biology, Leibniz Institute of Plant Biochemistry, 06120 Halle, Germany; gerd.balke@ipb-halle.de; 8Department of Pharmaceutical Chemistry and Bioanalytics, Institute of Pharmacy, Martin-Luther University Halle-Wittenberg, 06120 Halle, Germany; christian.ihling@pharmazie.uni-halle.de; 9All-Russia Research Institute for Agricultural Microbiology, 196608 St. Petersburg, Russia; vladimir.zhukoff@gmail.com (V.A.Z.); arriam2008@yandex.ru (I.A.T.); 10Department of Genetics and Biotechnology, St. Petersburg State University, 199034 St. Petersburg, Russia

**Keywords:** advanced glycation end products (AGEs), drought, glycation, SH-SY5Y human neuroblastoma cells, metabolomics, osmotic stress, pea (*Pisum sativum* L.), pro-inflammatory, seeds, seed metabolism, signaling pathways

## Abstract

Protein glycation is usually referred to as an array of non-enzymatic post-translational modifications formed by reducing sugars and carbonyl products of their degradation. The resulting advanced glycation end products (AGEs) represent a heterogeneous group of covalent adducts, known for their pro-inflammatory effects in mammals, and impacting on pathogenesis of metabolic diseases and ageing. In plants, AGEs are the markers of tissue ageing and response to environmental stressors, the most prominent of which is drought. Although water deficit enhances protein glycation in leaves, its effect on seed glycation profiles is still unknown. Moreover, the effect of drought on biological activities of seed protein in mammalian systems is still unstudied with respect to glycation. Therefore, here we address the effects of a short-term drought on the patterns of seed protein-bound AGEs and accompanying alterations in pro-inflammatory properties of seed protein in the context of seed metabolome dynamics. A short-term drought, simulated as polyethylene glycol-induced osmotic stress and applied at the stage of seed filling, resulted in the dramatic suppression of primary seed metabolism, although the secondary metabolome was minimally affected. This was accompanied with significant suppression of NF-kB activation in human SH-SY5Y neuroblastoma cells after a treatment with protein hydrolyzates, isolated from the mature seeds of drought-treated plants. This effect could not be attributed to formation of known AGEs. Most likely, the prospective anti-inflammatory effect of short-term drought is related to antioxidant effect of unknown secondary metabolite protein adducts, or down-regulation of unknown plant-specific AGEs due to suppression of energy metabolism during seed filling.

## 1. Introduction

In the most general way, protein glycation can be defined as an array of non-enzymatic post-translational modifications, formed by interaction of N-terminus and/or side chains of nucleophylic residues with reducing sugars and carbonyl products of their degradation [1]. Reducing sugars, aldoses and ketoses, readily react with lysyl residues of proteins forming Schiff base adducts, which readily undergo further Amadori or Heyns rearrangements to yield keto- and aldoamines, respectively [2,3] (Figure 1). Carbohydrate derivatives—sugar phosphates [4,5], sugar acids [6] and nucleotides [7]—were also reported as glycation agents. The products of early glycation, also known as Amadori and Heyns compounds, can be involved in oxidative degradation, often referred to as glycoxidation [8], yielding a structurally diverse group of advanced glycation end products (AGEs) [9]. Alternatively, AGEs can be formed by so-called oxidative or autoxidative glycosylation, i.e., formation of α-dicarbonyl compounds, mostly glyoxal (GO), methylglyoxal (MGO) and 3-deoxyglucosone (3-DG), and their interaction with lysyl, arginyl and cysteinyl residues [10]. Thereby, monosaccharide autoxidation, i.e., metal-catalyzed oxidation of sugars [11], oxidation of Schiff bases (Namiki pathway) [12], non-oxidative pathway [13], lipid metabolism [14], and non-enzymatic conversion of glycolytic intermediates glyceraldehyde-3-phosphate and dihydroxyacetone-phosphate [15] are recognized as the major pathways of α-dicarbonyl formation. However, the polyol pathway [16], acetone and threonine metabolism [17] might impact on the α-dicarbonyl pool as well (Figure 1).

During the last six decades, the food chemistry and medicinal aspects of AGE formation were intensively elaborated. Indeed, on one hand, AGEs readily form during thermal processing of foods, often compromising their dietary properties [18]. On the other, in mammalian organisms, the accumulation of AGEs accompanies ageing and contributes diabetic complications [19], Alzheimer’s [20] and Parkinson’s diseases [21]. The adverse effects of AGEs in mammals are usually attributed to (*i*) changes in protein conformation due to side chain modifications, (*ii*) intra- and inter-molecular cross-linking, affecting mechanical properties of tissues, and (*iii*) interaction with cell receptors triggering inflammatory responses [22].

In contrast, only recently glycation was reported in plants [23], and its protein targets along the accompanying characteristic patterns of side chain modifications were characterized [24]. Thereby, in comparison to the mammalian proteins, plant polypeptides demonstrated much higher numbers of AGE sites, where modifications were formed mostly via oxidative glycosylation [25]. Moreover, a significant increase of glycation levels at specific protein residues was characteristic for ageing of Arabidopsis leaves [26] and pea root nodules [27]. Not less importantly, enhanced AGE formation accompanied environmental stress [28]. In particular, under drought conditions, increase of α-dicarbonyl production was related to accumulation of glyoxal-derived modifications [29].

Indeed, drought, usually defined as a period of below normal precipitation that limits plant productivity in natural or agricultural systems [30], results in dramatic decrease of water potential (ψ_w_) of plant tissues [31]. The resulted dehydration triggers fast plant responses, i.e., stomata closure, inhibition of CO_2_ assimilation, accumulation of reducing equivalents in cells, overload of electron transport chains and overproduction of reactive oxygen species (ROS) [32]. Metabolic adjustment, i.e., the accumulation of amino acids, polyamines, polyols and sugars to prevent dehydration by reducing tissue ψ_w_ values [33], represents the basic stress avoidance mechanism coming into play at the next step. However, the accumulation of reducing sugars at the background of high contents of transition metals and enhanced ROS production ultimately results in enhancement of metal-catalyzed monosaccharide autoxidation and generation of highly reactive α-dicarbonyl precursors of AGEs [11]. Indeed, multiple key metabolites of the primary plant metabolic pathways are known as efficient glycation agents or can readily yield pro-glycative compounds during their oxidative degradation [24]. Recently, we demonstrated that drought results in glycation at specific arginine and lysine sites in Arabidopsis leaf proteins, yielding mostly *N^ε^*-(carboxymethyl)lysine (CML), *N^δ^*-(carboxymethyl)arginine and glyoxal-derived hydroimidazolone (Glarg) [29]. Thus, it can be assumed, that even short-term drought, i.e., the conditions which are difficult to avoid in agricultural practice, might result not only in enhancement of protein glycation at specific sites, but also in the overall accumulation of AGEs in crop plants and also in filling seeds. Due to their pro-inflammatory effects [22], the accumulation of AGEs in the seeds of harvested drought-exposed crops might compromise the quality and nutritional properties of plant-derived foods.

Addressing protein glycation, one needs to keep in mind, however, that plant cells are rich in secondary metabolites, phenolics, terpenes and alkaloids, which need to be considered as potential protein modification agents as well [34,35,36]. Not less importantly, these secondary metabolites can interfere with glycation and oxidation, directly affecting protein modification rates [37]. Indeed, it is well known that the people consuming exclusively plant-derived foods (vegetarians) have increased AGE blood levels [38]. However, such individuals are less suffering from inflammation-related diseases like atherosclerosis and diabetes mellitus [39]. Thus, on one hand, protein glycation needs to be considered in the context of the seed metabolome dynamics. On the other, one needs to keep in mind that, due to high chemical diversity of plant metabolites, glycation and oxidation represent only one of multiple possible protein modifications in plants.

These considerations brought us to the conclusion, that mild transient drought conditions (which are difficult to avoid in field) can trigger metabolic alterations in seeds, which might either suppress glycation of seed proteins and/or affect their pro-inflammatory properties in mammals. Despite of the importance of this aspect, it was not appropriately studied so far. Therefore, here we address whether the oxidative stress and metabolic alterations, triggered by a short-term experimental drought, can result in accumulation of anti-nutritive protein-bound AGEs, potentially affecting biological properties of mature seeds and potentially triggering pro-inflammatory responses. As in mammals dietary glycation adducts are absorbed in the intestine in the form of amino acid derivatives or, as was proved for the most representative AGE-CML, dipeptides [40,41], our workflow relied on quantitative solubilization of the total protein fraction followed with its exhaustive enzymatic hydrolysis according to the well-established protocol of Glomb and co-workers [42]. Recently, we demonstrated the compatibility of this workflow with solubilization of proteins and developed an adequate purification technique to make hydrolyzates applicable to cell assays [43]. Thus, our experimental setup can be considered as a simplified model of food digestion *in vitro*. In this proof of concept experiment, we refused the comprehensive modeling of food digestion conditions, as described by Hellwig et al. [41], to preserve the patterns of pea seed AGEs as richly as possible. Having this methodology in hand, we employed SH-SY5Y human neuroblastoma cells to address the effect of drought on the inflammatory response, triggered by pea seed protein.

## 2. Results

### 2.1. Establishment of Drought Stress and Characterization of Plant Stress Response

All seeds of the pea cultivar SGE successfully germinated and all resulting seedlings were efficiently inoculated with the rhizobial culture (Appendix A). A short-term drought (simulated by PEG-induced osmotic stress) was applied to mature plants directly after formation of last pods, i.e., at the beginning of senescence onset. The series of preliminary experiments, relying on assessment of physiological stress markers (stomatal conductivity, chlorophyll content and photosystem II efficiency) and the PEG 8000 concentration range of 2.5–10.0% (*w*/*v*) showed that already the lowest PEG concentration resulted in a 2-fold decrease of stomatal conductivity (Appendix A), although the other markers remained unaffected by this stressor dosage (Appendix A). Based on this fact, the dehydration, triggered by 2.5% (*w*/*v*) PEG 8000 could be considered as minimal drought, and these treatment conditions were, therefore, used in all following experiments.

On the 46th day, all plants had normal turgor and color (Figure 2A,B). After application of a two-day experimental drought (osmotic stress), all PEG-treated plants demonstrated clear dehydration symptoms, i.e., loss of turgor in comparison to the control group, which, however, was not accompanied with loss of green color (Figure 2C,D). After the subsequent recovery of drought-treated plants, followed with seed maturation and drying of pods, no difference in appearance of the control and experimental plants could be observed (Figure 2E). In agreement with this, the seeds of the both groups had the same morphology and anatomy (Appendix A).

The plant dehydration (manifested as a 2.5-fold decrease of water potential ψ_w_, Figure 3A) was accompanied by a statistically significant decrease in the following key physiological parameters: leaf relative water content (LRWC), stomatal conductance, photosystem II (PSII) efficiency and chlorophyll content (Figure 3B–E). This was in agreement with a 12-fold stress-related increase in abscisic acid (ABA) contents (Figure 3F). The accompanying oxidative stress demonstrated low intensity: slight, but significant increase in the contents of thiobarbituric acid-reactive substances could be observed (Figure 3G), the tissue abundances of hydrogen peroxide, lipid hydroperoxides, ascorbate and dehydroascorbate (as well as the total ascorbate pool and ascorbate/dehydroascorbate ratio) remained unaffected (Figure 3H and Appendix A).

### 2.2. Assessment of Seed Quality

Although the number of pods on individual plants varied essentially within the groups, no differences between the control and drought-treated cohorts could be observed. Thereby, each pod contained 4–10 seeds. The average weight and length of the pods, as well as the numbers of seeds per pod and seed weights did not show any inter-group differences (Figure 4A), although the seeds of the stress-exposed group demonstrated significantly increased protein contents in comparison to the controls (Figure 4B). However, it did not affect the physiological quality of seeds (i.e., the parameter known as vigor) which was assessed by a seed germination test (Figure 4C). Germination started already on the second day, although the largest number of germinating seeds could be observed on the fourth day with more than 95% germination rate achieved on the fifth day. The seeds, obtained from the plants subjected to a short-term drought, demonstrated a slight non-significant germination delay during the first three days of the experiment.

### 2.3. Metabolomics Analysis of Pea Seeds

The analysis of the pea seed metabolome revealed in total 3307 features, 175 of which could be annotated as primary thermally stabile metabolites (analyzed by GC-MS in aqueous methanolic extracts), 126 represented primary anionic thermally labile metabolites (analyzed by IP-RP-UHPLC-MS/MS in acidified aqueous ethanolic extracts) and 3006 features were detected in the analysis of semi-polar secondary metabolites. Thereby, in total 129 analytes showed drought-related alterations of at least 1.5-fold (*p* < 0.05, Figure 5 and Appendix A). Among this number, 87 and 42 analytes represented the groups of primary and secondary metabolites, respectively. The FDR correction (Benjamini–Hochberg) at the confidence level of *p* < 0.1 resulted in 108 confidently regulated primary metabolites. Among this number, ten represented thermally stabile metoxymated and/or trimethylsilylated derivatives (GC-MS dataset), whereas 88 constituted anionic thermally labile compounds (IP-RP-UHPLC-MS/MS dataset, Appendix A, respectively). Based on the database search by spectral similarity using the NIST library, open access resources of GMD and HMDB and in-house libraries (Appendix A), 85 and five primary and secondary metabolites, respectively, could be identified whereas the other differentially abundant species were annotated as unknowns (Appendix A). The most of the metabolites demonstrated a dramatic abundance decrease (up to 25-fold) in the seeds of the drought-treated plants in comparison to the untreated controls, whereas the relative contents of only 15 compounds (homoserine, turanose, melibiose, three monosaccharides, one sugar acid, one oligosaccharide and six unknowns) increased.

Despite of an essential number of drought-regulated features, principal component analysis (PCA) clearly demonstrated that the control and drought-treated groups can be distinguished only at the level of the primary thermally stabile metabolites (Appendix A). Although most of the individual anionic thermally labile metabolites were differentially abundant at the confidence level of *p* < 0.05 (Appendix A), this group of compounds showed relatively high dispersion, which resulted in high *p* values, obtained after FDR correction. To address the changes in primary seed metabolome, related to the water stress in more detail, we characterized the effects of the PEG-induced dehydration on individual metabolic pathways. For this, the list of the identified primary metabolites (Appendix A) was submitted to the Pathway Analysis tool in terms of the Metaboanalyst platform. Thereby, annotation of biochemical processes relied on the Kyoto Encyclopedia of Genes and Genomes (KEGG) pathway library of the model plant *Arabidopsis thaliana* L. This analysis revealed 57 metabolic pathways affected by short-term drought (Figure 6, Table 1, and Appendix A). The drought-responsive pathways were ranked by (*i*) confidence (*p*-values) of the differences in metabolite abundances and (*ii*) pathway impact, i.e., a quantitative characteristic of the metabolite contribution in the pathway. Based on this approach, the primary metabolism pathways involved in the seed response to PEG-induced drought could be characterized. Thus, the metabolism of galactose, glycine, serine, threonine and pyrimidine, as well as isoquinoline alkaloid biosynthesis, were the principal contributors in the observed drought-related metabolic alterations.

As, no significant drought-related changes were observed in the patterns of seed secondary semi-polar metabolites after the FDR correction, we addressed the general tendencies in the secondary metabolome response to drought. To obtain this information, the original data matrixes representing integrated peak areas of all features registered in positive and negative modes (1092 and 207 MS^1^ signals, respectively) were analyzed by partial least squares discriminant analysis (PLS-DA). The features with intensity threshold > 5000 counts and demonstrating at least 1.5-fold drought-related alterations in intensity, were additionally subjected to hierarchical clustering analysis (HCA) using the MetFamily software tool. It gave us access to individual metabolite groups, which could be assigned by characteristic patterns of fragment signals in tandem mass spectra. The characteristic fragment signals attributed to compounds of certain chemical classes were selected from predicted tandem mass-spectral library deposited in Human Metabolome Database (HMDB, http://www.hmdb.ca/). A search for specific diagnostic fragment ions, characteristic for different groups of metabolites revealed several chemical families, six of which could be identified—coumarine derivates (97), metabolites with phosphocholine group (527), oxylipins (26), metabolites conjugated with sugar moieties (2), fatty acid residues (63), metabolites with phosphate group (49), which could be visualized in PLS-DA and HCA models. The results clearly indicated that metabolites, featured with a phosphocholine group (most probably phosphatidylcholines), and oxylipins are more represented in the seeds of the drought-treated plants in comparison to those from controls. In contrast, coumarine derivates and sugar conjugates dominated in the extracts, obtained from the seeds of untreated plants (Appendix A).

### 2.4. Protein Extraction and Exhaustive Enzymatic Hydrolysis

The total protein fraction was successfully extracted from ground seed material. After reconstitution in aq. 10% (*w*/*v*) SDS solution, the protein concentrations and yields were in the range 26.6–44.8 mg/mL and 78.4–110.3 mg/g fresh weight, respectively (Appendix A). The assay precision was confirmed by SDS-PAGE loading 10 µg of protein on each lane: the overall lane densities were 3.7×10^4^±1.87×10^3^ arbitrary units (AU, RSD=5.06%, Appendix A).

The protein samples dissolved in aq. 10% (*w*/*v*) SDS were successfully hydrolyzed by the protocol of Glomb and co-workers [42] after a 20-fold dilution with phosphate buffer saline. The completeness of the enzymatic hydrolysis was confirmed by SDS-PAGE (Appendix A) with 30 mg aliquots of hydrolyzates. The protein digestion was considered to be complete, as the bands of the major pea storage proteins legumin (α- and β- subunits, ∼40 kDa and ∼20 kDa, correspondingly), vicilin (subunits of ∼29 kDa, ∼35 kDa and ∼47 kDa) and convicilin (subunit of ∼71 kDa), were not detectable. This conclusion was based on the assumption that staining sensitivity was better than 30 ng [25] and a legumin content of at least 80% of the total seed protein [44]

### 2.5. In Vitro Biological Effects of Pea Seed Proteins

To address the possible biological effects of the pea seed protein, related to the observed drought-related changes in seed metabolism, we assessed activation of phosphoinositide 3-kinases/RAC-alpha serine/threonine-protein kinase/mammalian target of rapamycin (PI3K/Akt/mTOR), mitogen-activated protein kinase/extracellular signal-regulated kinases (MAPK/ERK) and Janus kinase/signal transducer and activator of transcription (JAK/STAT) signaling pathways in mammalian cells by seed protein hydrolyzates, obtained from drought-treated plants in comparison to those isolated from the control ones. Thereby, activation of individual pathways was quantitatively assessed in human neuroblastoma cell culture SH-SY5Y by the degree of phosphorylation at specific sites of the characteristic marker proteins, visualized by Luminex^®^ xMAP^®^ technology.

As, based on our earlier results on drought-related enhancement of AGE production in Arabidopsis leaf [29] and known biological activities of AGEs [45], we expected an increase of pro-inflammatory properties of seed proteins upon the drought treatment. Therefore, the inflammation-inducing canonical NF-κB pathway was addressed first. The amounts of the hydrolyzates, supplemented to the culture medium, and treatment times were optimized in the series of preliminary experiments (Appendix A). Thus, the first round of optimization showed, that phosphorylation levels of all analyzed signaling proteins—IκB, FADD, IKK α/β, TNFR1, NF-κB (components of the canonical activation pathway of NF-κB) and c-myc (considered as one of the transcriptional target of NF-κB)—was significantly suppressed after 0.5 h treatment with 2.5 and 5 mg/mL hydrolyzates in comparison to lower supplementation amounts (1 mg/mL, Appendix A). On the other hand, 0.0625–1 mg/mL protein hydrolyzates, obtained from control seeds, did not show any statistically significant differences in cell response in comparison to the cells, treated with hydrolyzate-free medium for the same exposure time (0.5 h, Appendix A). Therefore, the hydrolyzate concentrations of 0.5 and 1 mg/mL were used for the optimization of treatment times. However, as was shown in the next optimization round, longer incubation times (3 h) resulted in a statistically significant increase of NF-κB and IκB phosphorylation levels in cells, treated with protein hydrolyzates from control pea seeds in comparison to the medium-treated cells (Appendix A). Based on these results, in further experiments, protein hydrolyzates were supplemented to the culture medium at the concentration of 0.5 mg/mL for 3 h.

Accordingly, a 3 h treatment of SH-SY5Y with 0.5 mg/mL protein hydrolyzates, obtained from the seeds of both control and drought-treated pea plants, resulted in the increase of the IκB, TNFR1 phosphorylation levels in comparison to those in medium-treated cells, whereas for c-myc and IKK α/β phosphorylation was down-regulated (Figure 7). However, surprisingly, the phosphorylation levels of IκB, FADD, IKK α/β and NF-κB were decreased in the cells, treated with hydrolyzates from the seeds of drought-exposed plants in comparison to the treatment with the hydrolyzates isolated from the control ones.

At the next step, the key players of several other signaling pathways impacting on NF-κB activation were addressed by the same approach. Thus, the treatment of SH-SY5Y cells with 0.5 mg/mL pea seed protein hydrolysates of both untreated and drought-treated plants for 3 h resulted in increased phosphorylation of ERK 1/2, Akt, JNK, CREB, and p38. Thereby, the hydrolyzates from the seeds of drought-exposed plants resulted in lower phosphorylation response of p38, JNK, NF-κB, p70S6K, STAT3, STAT5 compared to hydrolyzates of the control plants (Figure 8).

### 2.6. Protein-Bound Glycation Adducts in Pea Seeds

To address the question, if the observed suppression of signaling pathway activation was related to seed protein glycation, prospectively accompanying the plant response to a short-term drought in maturating pea seeds, the patterns of AGE-modified amino acids were analyzed by liquid chromatography-tandem mass spectrometry (LC-MS/MS). Statistical analysis revealed no significant differences between the control and drought-treated groups (Figure 9), although a trend towards an increase in the content of *N^δ^*-(2-hydro-5-methyl-4-imidazolon-2-yl)ornithine, *N^ε^*-(lactoyl)lysine, *N^δ^*-(carboxymethyl)arginine and *N^ε^*-(carboxyethyl)lysine after treating plants with PEG-8000 solution, could be observed. Thus, a short-term minimal drought did not cause accumulation of AGEs, which is generally consistent with the decrease in the contents of the major sugars (Figure 5) and a low intensity of the oxidative stress, developing in response to dehydration (Figure 3G,H).

## 3. Discussion

### 3.1. Effect of Experimental Drought on Pea Seed Metabolome

Obviously, all drought-related changes in the patterns of protein-bound AGEs and biological activities of isolated protein preparations can be attributed to alterations in plant metabolome and redox status. Due to stomata closure and overload of electron transport chains, dehydration results in pronounced suppression of the energy metabolism—tricarboxylic acid (TCA) cycle, pentosophosphate pathway and Calvin cycle—although glycolysis is usually enhanced under drought conditions [46,47]. On the other hand, persisting drought shifts the plant stress response to the dehydration avoidance strategy, typically manifested as mobilization of plant antioxidant systems in parallel to accumulation of amino acids and several sugar-related metabolites, usually referred to as metabolic (osmotic) adjustment [47,48]. Thus, a short-term water deficit would trigger stomata closure and oxidative stress, without metabolic adjustment, i.e., changes at the level of plant metabolome.

Our observations, in general, were in agreement with this scenario. Indeed, on one hand, dehydration resulted in a significant increase in the leaf contents of abscisic acid (ABA, Figure 3F), which triggered stomata closure followed with suppression of photosystem II activity (PSII) and decrease of chlorophyll contents (Figure 3C–E), that in turn resulted in development of oxidative stress (Figure 3G). On the other, the observed increase in reactive oxygen species (ROS) generation appeared to be relatively low and did not trigger any alterations in the status of ascorbate/dehydroascorbate system in leaves (Figure 3H). Thus, based on these data, no dramatic changes in the physiological response of mature seeds could be expected. Indeed, the applied short-term experimental drought did not affect seed and pod metrics and only minimally non-significantly delayed early steps (Figure 4).

However, despite the absence of visible drought-related alterations in seed quality, even a short-term osmotic stress dramatically affected biochemical status of the seeds. Surprisingly, the central metabolic pathways (glycolysis, citrate cycle, pentose phosphate cycle) were distinctly suppressed in seeds (Figure 6 and Table 1). Indeed, drought-treated plants demonstrated significantly higher protein yields in comparison to corresponding controls (106.9 ± 3.4 vs. 91.1 ± 12.6 mg/g fresh weight, Figure 4B.) without any changes in seed water contents. Recently, Hatzig et al. reported a significant increase (up to 30%) of protein contents in oilseed rape seeds, obtained from plants, subjected to the drought conditions during the whole period of embryo formation and development [49]. It can be assumed that the same mechanisms might underlie the effects, observed in the short-term osmotic stress treatment, described here. Accumulation of the seed reserve protein under drought conditions is in agreement with a strong down-regulation of the principal primary metabolites in mature seeds of drought-treated pea plants (Figure 5). Indeed, the pathway analysis revealed significant suppression of phenylalanine, tyrosine, tryptophan, histidine, arginine, proline, glycine and serine biosynthesis pathways (Figure 6 and Table 1). Thus, it seems to be likely, that under mild stress conditions, the plant mobilizes small molecules (amino acids and carbohydrates) as reserve substances (proteins and polysaccharides). The effect on polysaccharide metabolism can be illustrated by changes in starch, sucrose, fructose, mannose and amino sugar metabolism, as was illustrated by pathway analysis (Figure 6 and Table 1). In contrast, prolonged drought results in the seed metabolic changes, similar to those in leaf, i.e., in suppression of protein and lipid biosynthesis in parallel to up-regulation of sugar metabolism [50].

Thus, most likely, the minimal drought stress, applied here, could act as a priming factor, resulting in enhanced consumption of seed metabolic resources, amino acid building blocks for protein biosynthesis, carbohydrates as the energy source, nucleotides and co-enzymes as the substrates for the cellular energy generating machinery. These changes in accumulation of reserve substances might indicate adaptive response of plants to drought, which might be dependent on responsivity of pea cultivars to rhizobial symbiont [44]. Functional dissection of the seed primary metabolism (Figure 6 and Table 1) supports this conclusion: the pathways, related to the amino acid (glycine, serine, threonine, lysine, histidine, tryptophan, arginine, proline) and sugar (galactose, fructose, mannose) metabolism turned to be the most strongly down-regulated after the application of osmotic stress during seed filling. It can be expected that longer times (up to 1 week to 10 days) of drought application might result in more pronounced changes in metabolism of mature seeds. Thus, a pronounced decrease in protein and lipid contents, accompanied with increase in sugar contents, can be expected [50]. As was shown for Arabidopsis leaf, these changes can be accompanied with increase of glycation levels [28,29]. However, for mature seeds this aspect is still to be addressed in future studies.

### 3.2. Short-Term Drought Modulates Activation of NF-kB in SH-SY5Y Human Neuroblastoma Cells by Seed Protein Hydrolyzates

Plant protein hydrolyzates represent a complex mixture of amino acids and their natural derivatives. Thus, besides well-characterized glycated and oxidized amino acids [23], such hydrolyzates can contain adducts originating from oxidized fatty acids [51], i.e., ALEs [52]. Plant secondary metabolites, for example, phenolics, can form protein adducts as well: during the last decades it was shown for ferulic acid, hydroxytyrosol [53], epigallocatechin [54], rosmarinic acid [55] and many other representatives of this structural group. Moreover, the possibility of protein adduct formation needs to be considered for antraquinons [56] and sterols as well [57]. The proteins adducts, formed by plant natural products, are known for their antioxidant, antiproliferative, anti-diabetic effects and increase activity of α-amylase, trypsin, pepsin, lipase and lysozyme [34]. Thus, the overall effect of protein plant hydrolyzates on mammalian cells might be defined by a complex pattern of molecular interactions between cellular receptor structures and different adducts.

In mammals, AGEs are known to trigger inflammatory response via activation of multiple signaling pathways mediated by a broad range of cell surface (membrane-bound) and soluble receptors [58]. The primary target of AGEs at the membrane of mammalian cells is the receptor to advanced glycation end products (RAGE) [59], interaction with which is currently recognized as the main mechanism underlying AGE-induced inflammatory response [60]. The human SH-SY5Y cells also express this receptor, and often used to probe pro-and anti-inflammatory effects, also in the context of the Maillard reaction products [61,62,63]. Therefore, here we used this cell line to characterize prospective pro-inflammatory effects of AGEs, bound to the proteins, isolated from the seeds of drought-treated pea plants. Thereby, we addressed the main signaling cascades—PI3K/Akt, MAPK/ERK and JAK/STAT pathways—which result in activation of the transcription factors NF-κB, STAT3, and AP-1 (Figure 10) leading to the development of an inflammatory response [64].

The multi-target immunofluorescence approach allowed monitoring the phosphorylation status of all principal key-players of all three signaling pathways—transcription factors NF-κB and STAT3 and site-specific protein kinases ERK 1/2, p38, Akt, JNK, CREB, p70s6k and STAT5 (Figure 8). Treatment with the hydrolyzates, obtained from the seeds of control plants resulted in increase in IκB, TNFR1, ERK 1/2, Akt, JNK, CREB and p38 phosphorylation in comparison to medium-treated controls, that indicated activation of PI3K/Akt and MAPK/ERK pathways. However, despite of the expected drought-related up-regulation in phosphorylation of all kinases and factors, the key-players of the all three pathways (NF-kB, JNK and STAT3) showed decreased phosphorylation levels upon drought treatment (Figure 8). It indicated suppression of NF-kB, STAT3 and AP-1 phosphorylation and nuclear translocation by short-term drought. This was in agreement with the data on detailed dissection of the NF-kB pathway, indicating suppression of its activation at several steps in comparison to untreated controls (Figure 7).

This suppression of NF-kB activation cannot be potentially explained by any alteration in the patterns of known protein-bound AGEs, as their levels were not affected by short-term drought (Figure 9). It can be proposed that some unknown secondary metabolites, up-regulated under stress conditions (Appendix A), could form protein adducts with antioxidant properties, that is described in literature [65]. However, no significant drought-related abundance alteration could be revealed in this group after FDR correction. Therefore, we did not address this possibility in more detail and skipped structure assignment of differentially regulated secondary metabolites. However, formation of protein adducts of these metabolites cannot be excluded, and their crosstalk with AGEs still can be assumed. For example, CREB was found phosphorylated after treatment with pea protein hydrolyzates and it was able to inhibit NF-κB activation [66]. To clear this possibility, additional experiments are necessary.

To our opinion, the observed changes in the profiles of biological activity can be explained by drought-related alterations in the patterns of primary metabolites. The observed drought-related decrease in the seed contents of carbohydrates, sugar phosphates and nucleotides (Figure 5) could result in decrease of plant protein glycation. Remarkably, as the contents of the marker AGEs were not affected by drought (Figure 9), decrease in formation of some unknown novel AGEs can be proposed. Taking into account the formation pathways of the analyzed AGEs [67,68], it can be assumed, that glyoxal and methylglyoxal were not involved in their formation. It cannot also be excluded, that the Maillard reaction products, formed under drought condition have mostly antioxidant, but not pro-inflammatory properties [69], although the metabolic background does not support this assumption.

## 4. Materials and Methods

### 4.1. Reagents, Plant Material and Rhizobial Culture

Unless stated otherwise, materials were obtained from the following manufacturers: AMRESCO LLC (Fountain Parkway, Solon, OH, USA): ammonium persulfate (ACS grade), glycine (biotechnology grade), tris(hydroxymethyl)aminomethane (tris, ultra pure grade); Bio-Rad (Bio-Rad Laboratories GmbH, Feldkirchen, Germany): DC™ Protein Assay; Carl Roth GmbH and Co (Karlsruhe, Germany): ammonia solution (25%); sodium dodecyl sulfate (SDS, >99%), sodium chloride (p.a.), sodium phosphate dibasic dihydrate (p.a.), polyethylene glycol (PEG) 8000 (p.a.), 2-(*N*-morpholino)ethansulfonic acid (MES, p.a.), tris-(2-carboxyethyl)phosphine hydrochloride (TCEP, ≥98%), MgSO_4_x7H_2_O (extra pure), K_2_HPO_4_ (ACS grade); GE Healthcare (Munich, Germany): 2D Quant Kit; Macherey-Nagel GmbH and Co KG (Düren, Germany): *N*-methyl-*N*-(trimethylsilyl)trifluoroacetamide (MSTFA, MS grade), Chromabond C_18_ec polypropylene cartridges, 3 mL; PanReac AppliChem (Darmstadt, Germany): glycerol (ACS grade), Ca_3_(PO_4_)_2_ (pure, pharma grade); Reachem (Moscow, Russia): hydrochloric acid (p.a.), isopropanol (reagent grade); Roche (Basel, Switzerland): complete Mini Protease Inhibitor Cocktail; SERVA Electrophoresis GmbH (Heidelberg, Germany): acrylamide/bis-acrylamide solution (37.5/1, 30% (*w*/*v*), 2.6% C), Coomassie Brilliant Blue G-250, 2-mercaptoethanol (research grade), NB sequencing grade modified trypsin from porcine pancreas; Thermo Fisher Scientific (Bremen, Germany): PageRuler™ Prestained Protein Ladder #26616 (10–180 kDa), Dulbecco’s modified Eagle’s medium: Ham’s F12 medium 1:1 (DMEM/F12), Fetal Bovine Serum (FBS); VWR International (Leuven, Belgium): CoCl_2_×6H_2_O, FeSO_4_×7H_2_O (both ACS grade). Carboxypeptidase Y was isolated from yeast Saccharomyces cerevisiae as described by Johansen et al. [70] with minimal optimization (Appendix A). All other chemicals were purchased from Merck KGaA (Darmstadt, Germany). Water was purified in house on a water conditioning and purification system Millipore Milli-Q Gradient A10 system (resistance 18 mΩ/cm, Merck Millipore, Darmstadt, Germany).

Pea seeds of the cultivar SGE and rhizobial culture (*Rizobium leguminosarum* bv. viciae CIAM 1026) were provided by the All-Russia Research Institute of Agricultural Microbiology (St. Petersburg, Russia).

### 4.2. Plant Experiments

Pea seeds were stratified at 4°C for two days, germinated during two days in dark, transferred to vermiculite filled in 1 L pots and inoculated with rhizobial culture (*Rizobium leguminosarum* bv. viciae CIAM 1026). The plants were grown at 16 h light/8 h dark regimen at 21°C under 75% relative humidity. On the 42^nd^day after inoculation (d.a.i.) the plants were transferred to aerated aqueous medium, and five days later the medium was replaced with polyethylene glycol 8000 solution (PEG 8000) [71]. After two days of treatment, the plants were transferred to PEG-free aqueous medium, and later to soil. Afterwards, the plants were grown till the end of seed maturation. Plant shoots (*n* = 5, at least 2 g per biological replicate) were harvested directly before the transfer to the PEG-free medium, frozen in liquid nitrogen and ground in a Mixer Mill MM 400 ball mill with a 20 mm stainless steel ball (Retsch, Haan, Germany) at a vibration frequency of 30 Hz for 1 min. The ground material was stored at −80 °C till analysis. The mature seeds were harvested on the 66^th^ d.a.i., the numbers of pods per plant and the numbers of seeds in each pod were registered. Additionally, the weight and length of each pod, as well as seed weights were determined. After harvesting, seeds were left at room temperature for four days and stored at 4 °C afterwards.

### 4.3. Determination of Water Potential

Leaf water potential (ψ_w_) was determined by the gravimetric method as described by Paudel et al. [29] with minimal changes (*n* = 4). In detail, individual leaves were weighed and immediately immersed in 4.5 mL of serially diluted sucrose solutions (0.2–1 mol L^–1^) filled in Petri dishes (Ø 2 cm) covered with glass slides. After 2 h incubation at room temperature, the leaves were blotted carefully, weighed again, and the weight loss or gain was documented. The ψ_w_ values were calculated by the Vant Hoff equation (ψ_w_ = −*i*MRT), where M denotes the sucrose molar concentration yielding no changes in leaf weight upon a 2 h treatment.

### 4.4. Physiological and Biochemical Assays

Stomatal conductivity was assessed with a portable porometer(AP4, Delta-T Devices Ltd., Cottbus, Germany), whereas photosystem II (PS II) efficiency (variable fluorescence (Fv)/maximal fluorescence (Fm)) was determined using MINI-PAM-IIB fluorometer (Heinz Walz GmbH, Effeltrich, Germany), and relative chlorophyll content was measured using a diffusion chlorophyll meter SPAD-502 (Konica Minolta, Langenhagen, Germany) according to established protocols (for all experiments: *n* = 5, four-six leaves per plant) [72,73]. The leaf relative water content (LRWC) was calculated based on the difference of fresh and dry weight (3 d at 80°C) using the equation: LRWC (%)=(fresh weight–dry weight)x100%/fresh weight (*n* = 5, ten leaves per plant). For the determination of this parameter, 10 leaves from 5 control and 10 leaves from 5 drought-treated plants were used.

The assessment of biochemical stress markers relied on the methods, described by Frolov and co-workers [29,31]. In particular, the contents of hydrogen peroxide (Appendix A) and lipid hydroperoxides (Protocol S1-4) were quantified by oxidation of the Fe (II)xylenol orange complex. Lipid peroxidation products were determined as malondialdehyde (MDA) equivalents by the thiobarbituric acid (TBA) method (Protocol S1-5), whereas total ascorbate, ascorbic acid and dehydroascorbate contents were assessed by ascorbate oxidase method (Protocol S1-6). Determination of leaf abscisic acid (ABA) contents relied on the procedure of Balcke et al. [74] and ACQUITY H-Class UPLC ultrahigh performance liquid chromatography system (Waters GmbH, Eschborn, Germany) coupled on line to a QTRAP 6500 (AB Sciex, Darmstadt, Germany) triple quadrupole-linear ion trap instrument operating in negative multiple reaction monitoring (MRM) mode under the settings summarized in Appendix A.

### 4.5. Assessment of Seed Quality

Assessment of seed quality relied on the standards of the International Seed Testing Association, i.e., comprised germination [75]. For germination tests, the seeds of experimental and control groups (*n* = 4 × 14) were germinated eight days on wet filter paper in a climate chamber at 22°C. On the eighth day, the number of normally and abnormally (i.e., those with short, curved, or abnormally thick hypocotyls/roots) developed seedlings was assessed.

### 4.6. Metabolite Analysis

Analysis of temperature-stable polar primary metabolites relied on water-methanol extraction of frozen ground seed material (50 ± 10 mg) as described by Chantseva et al. [76] with minor changes. In detail, aliquots (8µL) of the extracts (a total extract volume was 1100 µL) were freeze-dried overnight. The residues were sequentially derivatized with methoxyamine hydrochloride (MOA) and *N*-methyl-*N*-(trimethylsilyl)trifluoroacetamide (MSTFA) according to the earlier established procedure [1]. The samples (1 µL) were analyzed by gas chromatography-electron ionization-quadrupole-mass spectrometry (GC-EI-Q-MS) using a GC2010 gas chromatograph coupled online to a quadrupole mass selective detector Shimadzu GCMS QP2010, equipped with a CTC GC PAL Liquid Injector (Shimadzu Deutschland GmbH, Duisburg, Germany) under the instrumental settings summarized in Appendix A.

Analysis of temperature-labile anionic primary metabolites and semi-polar secondary compounds relied on the three-step extraction procedure described by Balcke et al. [77] with minor modifications. In detail, approximately 150 mg of frozen seed material were placed in 2 mL polypropylene screw cap microtubes filled with 200 mg glass beads (0.75–1 mm diameter), 3 and 1 stainless steel beads of 3 mm and 5 mm diameter, respectively and supplemented with 900 µL of cold (−80 °C) dichloromethane (DCM)/ethanol mixture (2:1, *v*/*v*) with addition of 100 µL of ice-cold 1:200 HCl: water. After intensive homogenization (FastPrep-24^TM^, MP Biomedicals, Eschwege, Germany), 5.0 m/s, 3 × 20 s) and centrifugation (4 °C, 10 000 g, 5 min), 300 µL of the polar supernatant fraction were transferred into new pre-cooled 1.5 mL polypropylene microtubes. The residues were supplemented with 50 µL of ice-cold 1:200 HCl: water, the samples were mixed and centrifuged as described above, before 120 µL of the polar supernatant fractions were combined with the first portions and filtered on a polyvinylidene fluoride (PVDF) membrane (0.2 µm pore size) by centrifugation at 1000 rpm (4 °C, 10 min). At least 150 µL of the filtrate were transferred into glass inserts of chromatographic vials, covered with membrane-secured plastic caps. The samples were analyzed by reversed phase-ion pair-ultrahigh performance liquid chromatography-electrospray ionization-triple quadrupole tandem mass spectrometry (RP-IP-UHPLC-ESI-QqQ-MS/MS) using Waters ACQUITY H-Class UPLC System (Waters GmbH, Eschborn, Germany), coupled on line to a AB Sciex QTRAP 6500 LC-MS/MS System (AB Sciex, Darmstadt, Germany) under the chromatographic and mass spectrometric settings summarized in Appendix A.

The non-polar supernatant fractions (700 µL) were transferred to new 2.0 mL polypropylene microtubes and residues were supplemented with 500 µL cold (−80 °C) tetrahydrofuran. After mixing and centrifugation as described above, the non-polar supernatant fraction (500 µL) was combined with the first portion and dried in a N_2_ stream (TurboVap LV, Biotage, Uppsala, Sweden). The residues were reconstituted in 180 µL of a water-methanol mixture (1: 3, *v*/*v*), vortexed (3000 g, 15 s), centrifuged (4 °C, 10,000 g, 5 min), and the supernatants were filtered as described above. The samples were analyzed by reversed phase-ultrahigh performance liquid chromatography-electrospray ionization-quadrupole-time-of-flight mass spectrometry (RP-UHPLC-ESI-QqTOF-MS) using Waters ACQUITY I-Class UPLC System (Waters GmbH, Eschborn, Germany), coupled on line to a SciexTripleTOF 6600 LC-MS System (AB Sciex, Darmstadt, Germany), operated in positive or negative sequential window acquisition of all theoretical mass spectra (SWATH) mode. The chromatographic and mass spectrometric settings are specified in Appendix A.

Data pre-processing and processing of GC-MS data relied on the Automated Mass Spectral Deconvolution and Identification System (AMDIS, version 2.66 from 08.08.2008, free available via www.amdis.net), Xcalibur^TM^(version 2.0.7) and LCquan^TM^ (version 2.5.6, TermoFisher Scientific Inc., Bremen, Germany), MSDial (version 3.12, free available via http://prime.psc.riken.jp/Metabolomics_Software/MS-DIAL/index2.html). For interpretation of the LC-MS data MSDial, MetFamily (version 1.0, free available via https://msbi.ipb-halle.de/MetFamilyDevel), PeakView^TM^ (version 2.2) and MultiQuan^TM^ (version 3.0.2) tools (AB Sciex, Darmstadt, Germany) were used. Thereby, metabolite identification relied on a broad panel of available spectral libraries—National Institute of Standards and Technology (NIST), Golm Metabolome Database (GMD), Human Metabolome Database (HMDB), MS/MS and electron ionization (EI)-MS spectra curated by RIKEN Center for Sustainable Resource Science (free available via http://prime.psc.riken.jp/Metabolomics_Software/MS-DIAL/) and in-house library (partly with Kovats retention time indices, calculated by retention times of alkane standards, Appendix A). Quantitation relied on integration of the corresponding extracted ion chromatograms (XICs, ± 0.5 Da) at specific retention times (t_R_). The processing and statistical interpretation of the acquired data relied on MetaboAnalyst 4.0 and/or MetFamily 1.0 online platforms (free available via www.metaboanalyst.ca [78] and https://msbi.ipb-halle.de/MetFamilyDevel/ [79], respectively). The pathway analysis was based on the Pathway Enrichment Analysis by GlobalTest method as described by Goeman et al. [80] and was done by MetaboAnalyst 4.0.

### 4.7. Protein Isolation

The total protein fraction was isolated by phenol extraction procedure [81] with some modifications. In detail, the frozen ground seed material (approximately 400 mg per replicate) in 15 mL polypropylene tubes, was transferred to ice bath, and three min later supplemented with 5.6 mL of cold (4 °C) phenol extraction buffer (0.7 mol/L sucrose, 0.1 mol/L KCl, 5 mmol/L ethylenediaminetetraacetic acid (EDTA), 2% (*v*/*v*)-mercaptoethanol and 1 mmol/L phenylmethylsulfonyl fluoride (PMSF) in 0.5 mol/L tris-HCl buffer, pH 7.5). The samples were vortexed for 30 s, and 5.6 mL of cold phenol (4 °C) saturated with 0.5 mol/L tris-HCl buffer (pH 7.5) were added. Afterwards, the samples were incubated on a vertical rotation platform (30 rpm, 30 min, 4 °C) and centrifuged (2500 g, 30 min, 4 °C). The upper phenolic phases were transferred to the new 15 mL polypropylene tubes and washed two times with equal volumes of the phenol extraction buffer with vortexing (30 s), rotation (30 rpm, 30 min, 4 °C) and centrifugation (2500 g, 15 min, 4 °C) after each buffer addition. Finally, the phenolic phases were transferred to new 15 mL polypropylene tubes, supplemented with five volumes of cold 0.1 mol/L ammonium acetate in methanol, and placed to −20 °C overnight. Next morning, the proteins were pelleted by centrifugation (15 min, 2500 g, 4 °C). The pellets were washed twice with two volumes of cold (−20 °C) methanol (compared to the phenol phase volume), and once with cold acetone (with centrifugation at each step –2500 g, 10 min, 4 °C). The final pellets were dried under air flow under the hood for 1 h, reconstituted in 1.5 mL of 10% (*w*/*v*) SDS, and protein contents were determined by 2D Quant Kit according to the producer instructions.

### 4.8. Exhaustive Enzymatic Hydrolysis

Isolated total seed protein (*n* = 3, 15 mg per replicate) was subjected to exhaustive enzymatic hydrolysis as described by Smuda et al. [42] with some modifications. In detail, the appropriate volumes of protein solutions in 10% (*w*/*v*) SDS were transferred to 15 mL polypropylene tubes, adjusted to 10 mL with phosphate buffered saline (PBS, 137 mmol/L NaCl, 2.7 mmol/L KCl, 10 mmol/L Na_2_HPO4, 1.8 mmol/L KH_2_PO4, pH 7.4), and supplemented with 30 μL of 1 mol/L CaCl_2_ solution and a small crystal of thymol. The proteins were digested under continuous shaking (450 rpm) in dark by sequential incubation with the following enzymes (per 1 mg substrate): 0.8 units of pronase E (24 h, 37 °C twice), 0.13 units of proteinase K (18 h, 37 °C) and 0.05 units of carboxypeptidase Y (24 h, 25 °C), respectively. The completeness of proteolysis was confirmed by polyacrylamide gel electrophoresis in presence of sodium dodecyl sulfate (SDS-PAGE) as described by Greifenhagen et al. [82] (Protocol S1-7). Afterwards, SDS was removed from the protein hydrolyzates by solid phase extraction as described by Antonova et al. [43] (Protocol S1–8).

### 4.9. Quantitative Determination of Glycated Adducts

Quantification of *N^ε^*-(fructosyl)lysine- and AGE-modified lysine and arginine adducts relied on the procedure of Glomb and co-workers [42] employing a standard addition approach and instrumental analysis by ion pair-reversed phase-high performance liquid chromatography coupled on-line to electrospray ionization-triple quadrupole tandem mass spectrometry (IP-RP-HPLC-ESI-QqQ-MS/MS).

### 4.10. Cell Culture and Treatment with Protein Hydrolysates

The SH-SY5Y human neuroblastoma cells were maintained in Dulbecco’s modified Eagle’s medium:Ham’s F12 medium (DMEM:F12) 1:1 supplemented with 10% (*v*/*v*) Fetal Bovine Serum (FBS), 100 U/mL penicillin, and 100 g/mL streptomycin, at 37 °C in a humidified atmosphere containing 5% CO_2_. Cells were grown to 80–90% confluence in 25 mm^2^ flasks and the medium was replaced every third day.

One day before the analysis, the cells were seeded into 24-well culture plates at the density of 1× 10^6^ cells per well. After removal of medium, the pea seed protein hydrolyzates, obtained from the drought-treated and control plants, were applied to the cells by supplementation to fresh FBS-free medium. Concentrations of the applied protein hydrolyzates and incubation times were optimized in preliminary experiments with protein hydrolysates isolated from control seeds. For this, the cells were exposed to different concentrations of pea hydrolysates (0.0625–5 mg/mL of total protein) for various treatment times (0.5–24 h). Culture medium without FBS served as control.

### 4.11. Analysis of Signaling Pathways

Cells, treated with hydrolyzates, were washed twice with Hanks’ Balansed Salt Solution, and lysed with a lysis buffer, supplemented with a protease inhibitor mix, phosphatase inhibitor cocktail 2 and benzonase, according to the manufacturer’s protocol. The protein concentrations in cell lyzates were determined by the Lowry method [83] and adjusted in all samples with the lysis buffer. Analysis of the proteins involved in NF-κB-mediated signaling—c-Myc, Fas-associated protein with death domain (FADD, Ser194), nuclear factor of kappa light polypeptide gene enhancer in B-cells inhibitor, alpha (IκBα, Ser32), inhibitor of nuclear factor kappa-B kinase subunits alpha/beta (IKKα/β, Ser177/Ser181), nuclear factor kappa-light-chain-enhancer of activated B cells (NF-κB, Ser536), Tumor necrosis factor receptor 1 (TNFR1), in lyzates relied on the Luminex^®^ xMAP^®^ multiparameter immunofluorescence technology (Bio-Plex 200, Bio-Rad) and MILLIPLEXMAP NF-κB Signaling 6-plex Magnetic Bead Kit 96-well Plate, applied according to the manufacturer’s instructions. The same cell lyzates were also analyzed with MILLIPLEX MAP Multi-Pathway Magnetic Bead 9-Plex Cell Signaling Multiplex Assay to evaluate the activation of other intracellular signaling pathways by the levels of phosphorylated extracellular signal-regulated kinases/mitogen-activated protein kinase 1/2 (ERK/MAP 1/2, Thr185/Tyr187), RAC-alpha serine/threonine-protein kinase (Akt, Ser473), signal transducer and activator of transcription 3 (STAT3, Ser727), c-Jun N-terminal kinases (JNK, Thr183/Tyr185), ribosomal protein S6 kinase (p70S6K, Thr412), NF-κB(Ser536), signal transducer and activator of transcription 5 (STAT5 A/B, Tyr694/699), cAMP response element-binding protein (CREB, Ser133), and p38 mitogen-activated protein kinase (p38, Thr180/Tyr182), after treatment with pea protein hydrolysates. The significance of the drought-related differences was assessed by one-way ANOVA with Tukey’s multiple comparisons test (*n* = 5, for five pea protein replicates, the total measurement number is 25).

## 5. Conclusions

Drought stress is accompanied with dehydration of plant tissues, which triggers stomata closure, development of oxidative stress and up-regulation of soluble carbohydrates and amino acids. High contents of reducing sugars at the background of ROS overproduction are favorable for protein glycation in plant leaves. However, the application of a short-term experimental drought at the stage of seed filling did not result in any alterations in the patterns of previously characterized glycation adducts in seed proteins. However, a two-day treatment with osmotic stress affected the biological activity of seed protein, i.e., reduced a potential of seed protein hydrolyzates to activate transcription factor NF-kB and trigger thereby pro-inflammatory response. This phenomenon can have several explanations: (*i*) drought-related formation of secondary metabolite adducts with antioxidant and/or antiglycative properties, (*ii*) formation of Maillard reaction products with antioxidant properties, and (*iii*) depletion of reactive primary metabolites under dehydration conditions. Under the observed metabolic background, the latter mechanism seems to be the most relevant. However, for complete understanding the underlying mechanism different drought models with different stressor dosages need to be addressed.

## Figures and Tables

**Figure 1 ijms-21-00567-f001:**
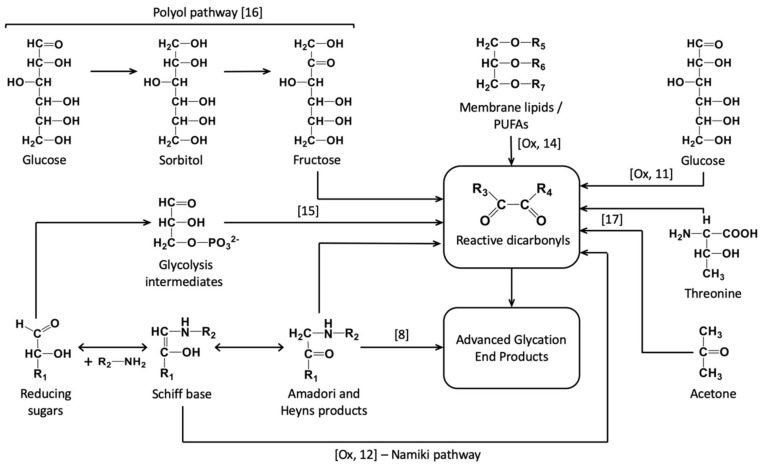
The main pathways of advanced glycation end product (AGE) formation in vivo: degradation of Amadori and Heyns products [8], lipid peroxidation [14], polyol pathway [16], monosaccharide autoxidation [11], oxidation of Schiff bases [12], non-enzymatic conversion of glycolysis intermediates [15], acetone and threonine metabolism [17]. [Ox], oxidation.

**Figure 2 ijms-21-00567-f002:**
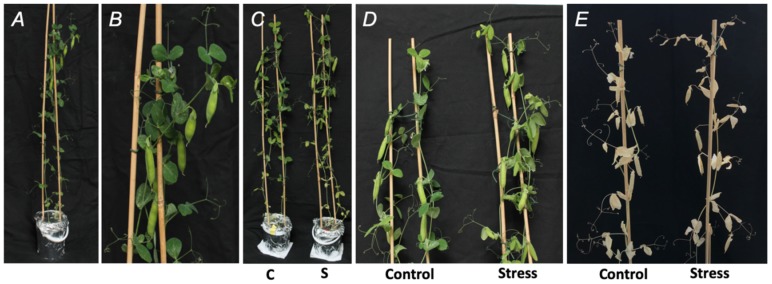
Pea (*Pisum sativum* L., cultivar SGE) plants before stress application (**A**,**B**), after a two-day exposure to the aqueous medium with and without supplementation of 2.5% (*w*/*v*) PEG 8000 (defined as Stress and Control, respectively, **C**,**D**), and after the completeness of seed maturation (**E**).

**Figure 3 ijms-21-00567-f003:**
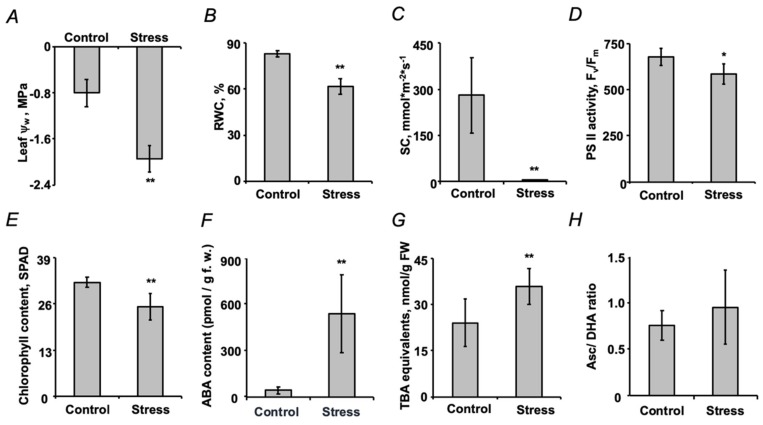
Characterization of plant stress after a two-day exposure of pea (*Pisum sativum* L., cultivar SGE) plants to the aqueous medium with and without supplementation of 2.5% (*w*/*v*) PEG 8000 (defined as stress and control, respectively) by water potential (**A**), leaf relative water content (LRWC, **B**), stomatal conductivity (**C**), photosystem II (PSII) efficiency (**D**), chlorophyll content (**E**), the contents of abscisic acid (ABA, **F**) and thiobarbituric acid-reactive substances (expressed as malondialdehyde equivalents, **G**), as well as ascorbate/dehydroascorbate ratio (**H**). ** and * denote statistical significance (*t*-test) at the confidence level *p* <0.01 and *p* < 0.05, respectively.

**Figure 4 ijms-21-00567-f004:**
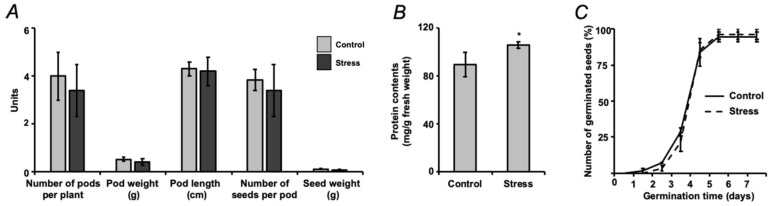
Characterization of pea (*Pisum sativum* L., cultivar SGE) seed quality, assessed after a two-day exposure to the aqueous medium with and without supplementation of 2.5% (*w*/*v*) PEG 8000 (defined as stress and control, respectively) during seed maturation: pod and seed metrics (**A**), protein contents (**B**) and germination rate (**C**); * denote statistical significance (*t*-test) at the confidence level *p* < 0.05.

**Figure 5 ijms-21-00567-f005:**
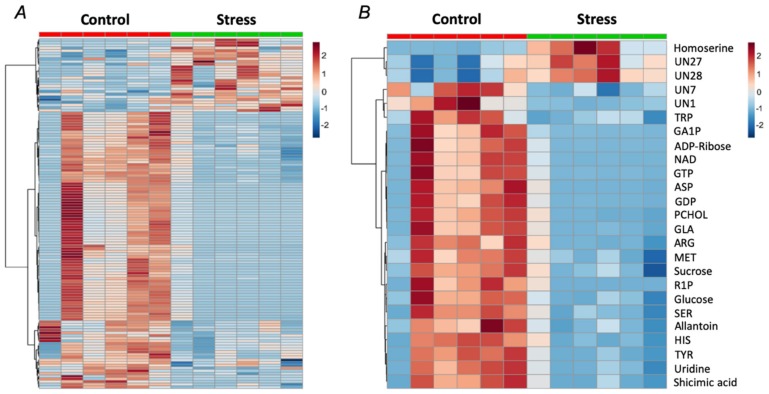
Heat map representing the features (both identified and unknown), differentially (more than 1.5-fold, *p* < 0.05) abundant in the mature pea (*Pisum sativum* L., cultivar SGE) seeds after a two-day exposure of the plants (at the seed maturation step) to the aqueous medium supplemented with 2.5% (*w*/*v*) PEG 8000 (stress, *n* = 6) in comparison to the seeds of the plants, treated with PEG-free medium (control, *n* = 6). The panel **A** represents the overall view of 129 features (listed in Appendix A), whereas the panel **B** represents the 20 most confident features (listed in the order of decreasing confidence, i.e., increasing *p* value), based on ANOVA. The non-identified features are labeled as unknowns (UN). UN 27, *m*/*z* 188.1648; UN 28, *m*/*z* 132.1023; UN 7, *m*/*z* 67.0297; UN 1, *m*/*z* 374; TRP, tryptophan; GA1P, glucosamine-1-phosphate; ADP-ribose, adenosine diphosphate ribose; NAD, nicotinamide adenine dinucleotide; GTP, guanosine-5′-triphosphate; ASP, aspartic acid; GDP, guanosine-5′-diphosphate; PCHOL, phosphocholine; GLA, glutamic acid; ARG, arginine; MET, methionine; R1P, ribose-1-phosphate; SER, serine; HIS, histidine; TYR, tyrosine. Heatmap was constructed in MetaboAnalyst 4.0 (https://www.metaboanalyst.ca).

**Figure 6 ijms-21-00567-f006:**
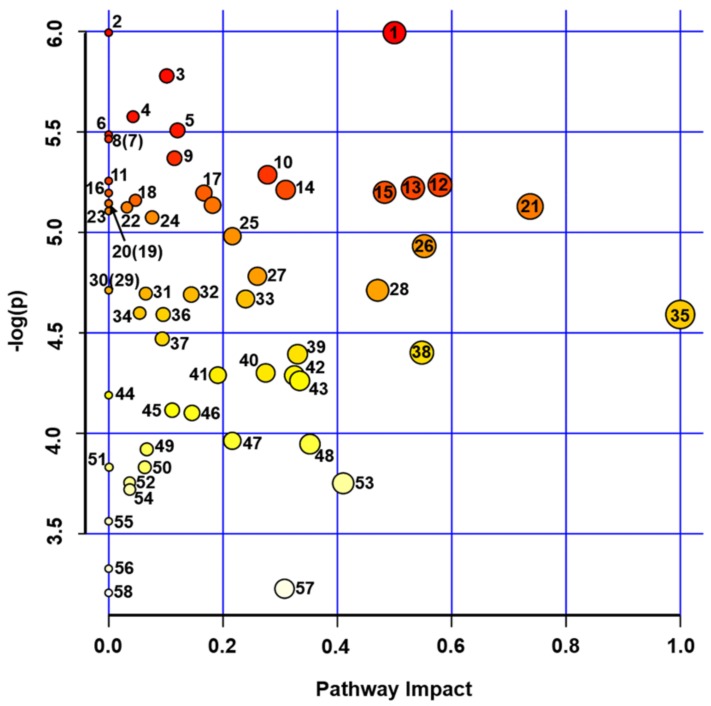
Pathway analysis accomplished for the drought-regulated (more than 1.5-fold, t-test *p* < 0.05, FDR adjusted at *p* < 0.1) primary polar metabolites of pea (*Pisum sativum* L., cultivar SGE) seeds annotated by untargeted GC-EI-Q-MS and targeted IP-RP-UHPLC-MS/MS. Annotation relied on authentic standards and in-house spectral library (Appendix A). To ensure the consistency of the sample numbers in the groups, the data for the stressed sample S12 was imputed via the random forest (RF) algorithm by the online tool MetImp. The pathway analysis combines the results from pathway enrichment analysis (global test) with pathway topology analysis (relative betweenness centrality) helping thereby to identify the most relevant pathways related to the stress conditions. The more intense red color of the circles indicates higher significance of the observed differences, whereas the circle size and position along the X-axis indicate the impact of the annotated metabolites in the corresponding pathways. The pathways (marked numerically) are listed in Table 1. For more information, please refer to Appendix A. To address the regulated metabolic pathways separately for the drought-regulated (more than 1.5-fold, t-test *p* < 0.05, FDR adjusted at *p* < 0.1) primary polar metabolites of pea (*Pisum sativum* L., cultivar SGE) seeds annotated by untargeted GC-EI-Q-MS and targeted IP-RP-UHPLC-MS/MS, please refer to Appendix A.

**Figure 7 ijms-21-00567-f007:**
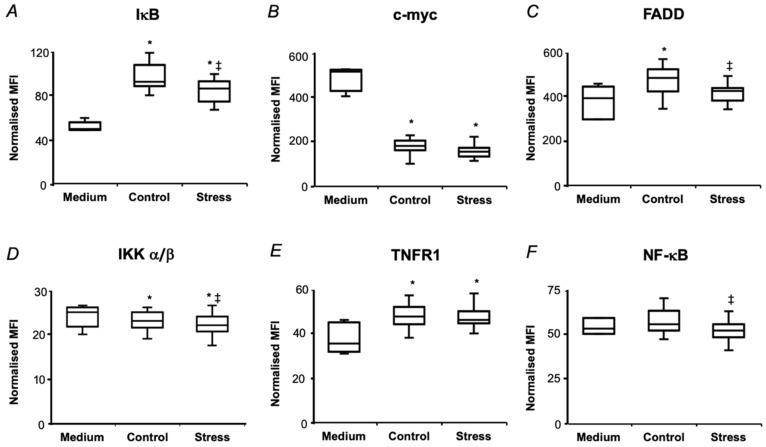
The effects of pea seed protein hydrolyzates on the NF-κB-mediated signaling pathway in SH-SY5Y human neuroblastoma cells. The total protein fraction was isolated from pea (*Pisum sativum* L., cultivar SGE) seeds after a two-day exposure of the plants (at the seed maturation step) to the aqueous medium with and without addition of 2.5% (*w*/*v*) PEG 8000 (defined as Stress and Control, respectively) and subjected to exhaustive enzymatic hydrolysis. The levels of phosphorylated IκB (**A**), c-myc (**B**), FADD (**C**), IKK α/β (**D**), TNFR1 (**E**), NF-κB (**F**) were determined in SH-SY5Y cell lyzates by Luminex^®^ xMAP^®^ technology after 3 h of incubation with protein hydrolysates (0.5 mg/mL), supplemented to the culture medium, and normalized to total protein content. The data are presented as median, inter-quartile range, minimal and maximal values, and were analyzed by one-way ANOVA with Tukey’s multiple comparisons test (*n* = 5, five technical replicates per biological one). Statistically significant differences (*p* < 0.05) in comparison to the medium-treated cells (Medium), to the cells, treated with protein hydrolyzates, obtained from the seeds of control plants (Control) are denoted as * and ‡, respectively. MFI, median fluorescent intensity.

**Figure 8 ijms-21-00567-f008:**
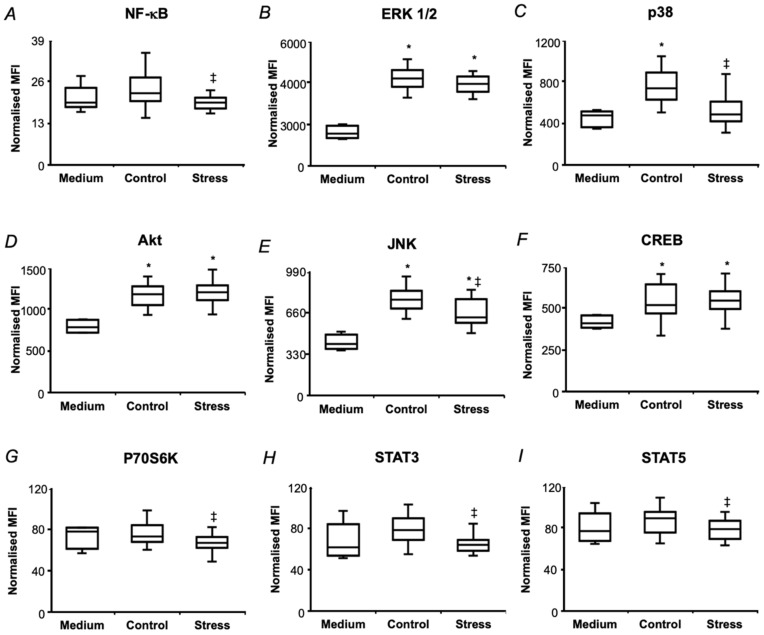
The effects of pea seed protein hydrolyzates on the MAPK/ERK, PI3K/Akt/mTOR and JAK/STAT signaling pathways in SH-SY5Y human neuroblastoma cells. The total protein fraction was isolated from pea (*Pisum sativum* L., cultivar SGE) seeds after a two-day exposure of the plants (at the seed maturation step) to the aqueous medium with and without addition of 2.5% (*w*/*v*) PEG 8000 (defined as Stress and Control, respectively) and subjected to exhaustive enzymatic hydrolysis. The levels of phosphorylated NF-κB (**A**), ERK 1/2 (**B**), p38 (**C**), Akt (**D**), JNK I (**E**), CREB (**F**), p70S6K (**G**), STAT3 (**H**), STAT5 (**I**) were determined in SH-SY5Y cell lyzates by Luminex^®^ xMAP^®^ technology after 3 h of incubation with protein hydrolysates (0.5 mg/mL), supplemented to the culture medium, and normalized to total protein content. The data are presented as median, inter-quartile range, minimal and maximal values, and were analyzed by one-way ANOVA with Tukey’s multiple comparisons test (*n* = 5, five technical replicates per biological one). Statistically significant differences (*p* < 0.05) in comparison to the medium-treated cells (Medium) and to the cells, treated with protein hydrolyzates, obtained from the seeds of control plants (Control) are denoted as * and ‡, respectively. MFI, median fluorescent intensity.

**Figure 9 ijms-21-00567-f009:**
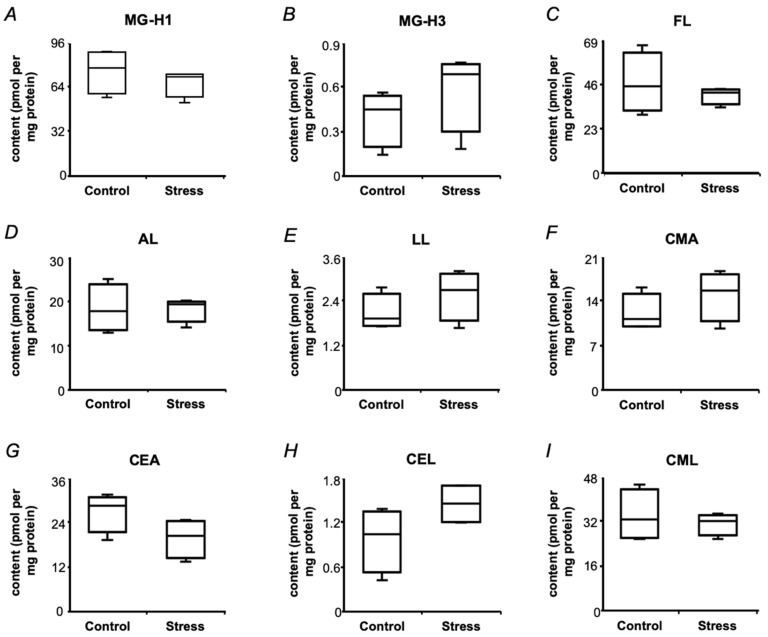
The contents of individual AGEs in mature pea (*Pisum sativum* L., cultivar SGE) seeds after a two-day exposure of plants (at the seed maturation step) to the aqueous medium with and without addition of 2.5% (*w*/*v*) PEG 8000 (defined as Stress and Control, respectively). The analysis relied on ion pair-reversed phase high-performance liquid chromatography coupled on-line to electrospray ionization-triple quadrupole tandem mass spectrometry (IP-RP-HPLC-ESI-QqQ-MS/MS). MG-H1, methylglyoxal-derived hydroimidazolone 1, *N^δ^*-(5-hydro-5-methyl-4-imidazolon-2-yl)ornithine (**A**); MG-H3, methylglyoxal-derived hydroimidazolone 3, *N^δ^*-(2-hydro-5-methyl-4-imidazolon-2-yl)ornithine (**B**); FL, *N^ε^*-(formyl)lysine (**C**); AL, *N^ε^*-(acetyl)lysine (**D**); LL, *N^ε^*-(lactoyl)lysiI (**E**); CMA, *N^δ^*-(carboxymethyl)arginine (**F**); CEA, *N^δ^*-(carboxyethyl)arginine (**G**); CEL, *N^ε^*-(carboxyethyl)lysine (**H**); CML, *N^ε^*-(carboxymethyl)lysine (**I**). The data are presented as median, interquartile range, minimal and maximal values (*n* = 5).

**Figure 10 ijms-21-00567-f010:**
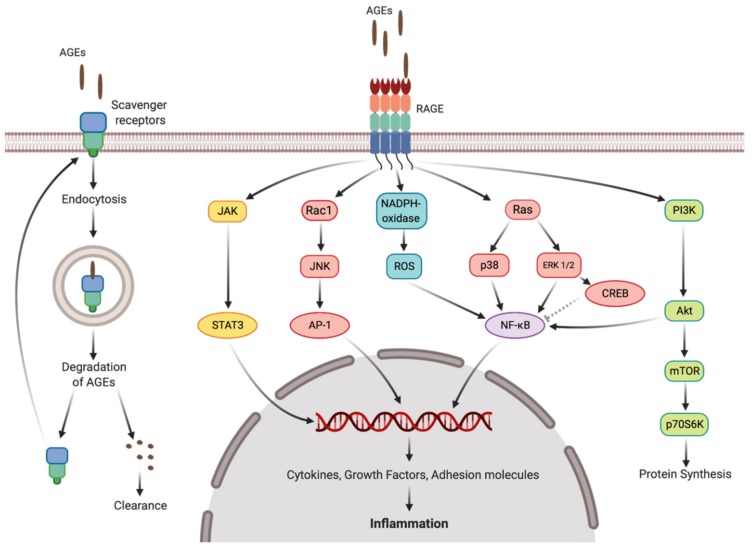
Advanced glycation end product (AGE)-mediated signaling events triggered by AGE-RAGE interaction and degradation of AGEs via lysosomal system. AGEs, advanced glycation end products; Akt, RAC-alpha serine/threonine-protein kinase; AP-1, activator protein 1; CREB, cAMP response element-binding protein; ERK 1/2, extracellular signal–regulated kinases 1/2; JAK, Janus kinase; JNK, c-Jun N-terminal kinases; mTOR, mammalian target of rapamycin; NF-B, nuclear factor kappa-light-chain-enhancer of activated B cells; p38, p38 mitogen-activated protein kinases; p70S6K, ribosomal protein S6 kinase; PI3K, phosphoinositide 3-kinases; RAGE, receptor for advanced glycation end products; ROS, reactive oxygen species; STAT3, signal transducer and activator of transcription 3.

**Table 1 ijms-21-00567-t001:** Metabolic pathways in mature seeds of pea (*Pisum sativum* L.) plants, subjected to short-term drought at the seed filling step.

# ^a^	Metabolic Pathway ^b^	Total Cmpd ^c^	Hits ^d^	Raw *p* ^e^	-LOG (*p*)	Holm Adjust ^f^	FDR ^g^	Impact ^h^
1	Isoquinoline alkaloid biosynthesis	6	1	2.49 × 10^−3^	5.99	1.47 × 10^−1^	1.48 × 10^−2^	0.5
2	Ubiquinone and other terpenoid-quinone biosynthesis	38	1	2.49 × 10^−3^	5.99	1.47 × 10^−1^	1.48 × 10^−2^	0
3	Phenylalanine, tyrosine and tryptophan biosynthesis	22	6	3.09 × 10^−3^	5.78	1.76 × 10^−1^	1.48 × 10^−2^	0.1
4	Histidine metabolism	15	2	3.79 × 10^−3^	5.58	2.12 × 10^−1^	1.48 × 10^−2^	0.04
5	Tryptophan metabolism	28	2	4.06 × 10^−3^	5.51	2.23 × 10^−1^	1.48 × 10^−2^	0.12
6	Indole alkaloid biosynthesis	4	1	4.14 × 10^−3^	5.49	2.24 × 10^−1^	1.48 × 10^−2^	0
7	Sphingolipid metabolism	17	1	4.24 × 10^−3^	5.46	2.25 × 10^−1^	1.48 × 10^−2^	0
8	Sulfur metabolism	15	1	4.24 × 10^−3^	5.46	2.25 × 10^−1^	1.48 × 10^−2^	0
9	Glycerophospholipid metabolism	37	2	4.66 × 10^−3^	5.37	2.38 × 10^−1^	1.48 × 10^−2^	0.12
10	Arginine and proline metabolism	34	3	5.06 × 10^−3^	5.29	2.53 × 10^−1^	1.48 × 10^−2^	0.28
11	Cyanoamino acid metabolism	29	5	5.22 × 10^−3^	5.26	2.56 × 10^−1^	1.48 × 10^−2^	0
12	Glycine, serine and threonine metabolism	33	6	5.32 × 10^−3^	5.24	2.56 × 10^−1^	1.48 × 10^−2^	0.58
13	Pyrimidine metabolism	38	16	5.41 × 10^−3^	5.22	2.56 × 10^−1^	1.48 × 10^−2^	0.53
14	Arginine biosynthesis	18	7	5.45 × 10^−3^	5.21	2.56 × 10^−1^	1.48 × 10^−2^	0.31
15	Nicotinate and nicotinamide metabolism	13	3	5.52 × 10^−3^	5.20	2.56 × 10^−1^	1.48 × 10^−2^	0.48
16	Monobactam biosynthesis	8	1	5.54 × 10^−3^	5.20	2.56 × 10^−1^	1.48 × 10^−2^	0
17	beta-Alanine metabolism	18	3	5.54 × 10^−3^	5.20	2.56 × 10^−1^	1.48 × 10^−2^	0.17
18	Carbon fixation in photosynthetic organisms	21	4	5.75 × 10^−3^	5.16	2.56 × 10^−1^	1.48 × 10^−2^	0.05
19	Porphyrin and chlorophyll metabolism	48	1	5.83 × 10^−3^	5.14	2.56 × 10^−1^	1.48 × 10^−2^	0
20	Nitrogen metabolism	12	2	5.84 × 10^−3^	5.14	2.56 × 10^−1^	1.48 × 10^−2^	0
21	Butanoate metabolism	17	4	5.89 × 10^−3^	5.13	2.56 × 10^−1^	1.48 × 10^−2^	0.18
22	Alanine, aspartate and glutamate metabolism	22	9	5.93 × 10^−3^	5.13	2.56 × 10^−1^	1.48 × 10^−2^	0.74
23	Cysteine and methionine metabolism	46	4	5.95 × 10^−3^	5.12	2.56 × 10^−1^	1.48 × 10^−2^	0.03
24	Glucosinolate biosynthesis	65	2	6.06 × 10^−3^	5.11	2.56 × 10^−1^	1.48 × 10^−2^	0
25	Folate biosynthesis	27	1	6.26 × 10^−3^	5.07	2.56 × 10^−1^	1.48 × 10^−2^	0.08
26	Ascorbate and aldarate metabolism	18	3	6.87 × 10^−3^	4.98	2.56 × 10^−1^	1.56 × 10^−2^	0.22
27	Glutathione metabolism	26	6	7.22 × 10^−3^	4.93	2.56 × 10^−1^	1.58 × 10^−2^	0.55
28	Terpenoid backbone biosynthesis	30	5	8.38 × 10^−3^	4.78	2.68 × 10^−1^	1.62 × 10^−2^	0.26
29	Phenylalanine metabolism	11	1	8.99 × 10^−3^	4.71	2.79 × 10^−1^	1.62 × 10^−2^	0.47
30	Phenylpropanoid biosynthesis	46	1	8.99 × 10^−3^	4.71	2.79 × 10^−1^	1.62 × 10^−2^	0
31	Tropane, piperidine and pyridine alkaloid biosynthesis	8	1	8.99 × 10^−3^	4.71	2.79 × 10^−1^	1.62 × 10^−2^	0
32	Glycerolipid metabolism	21	3	9.14 × 10^−3^	4.70	2.79 × 10^−1^	1.62 × 10^−2^	0.06
33	Galactose metabolism	27	8	9.19 × 10^−3^	4.69	2.79 × 10^−1^	1.62 × 10^−2^	0.14
34	Purine metabolism	63	14	9.39 × 10^−3^	4.67	2.79 × 10^−1^	1.62 × 10^−2^	0.24
35	Zeatin biosynthesis	21	5	1.01 × 10^−2^	4.60	2.79 × 10^−1^	1.62 × 10^−2^	0.05
36	Synthesis and degradation of ketone bodies	4	2	1.01 × 10^−2^	4.59	2.79 × 10^−1^	1.62 × 10^−2^	1
37	Valine, leucine and isoleucine degradation	37	2	1.01 × 10^−2^	4.59	2.79 × 10^−1^	1.62 × 10^−2^	0.1
38	Pentose and glucuronate interconversions	16	2	1.14 × 10^−2^	4.47	2.79 × 10^−1^	1.78 × 10^−2^	0.09
39	Starch and sucrose metabolism	22	6	1.23 × 10^−2^	4.40	2.79 × 10^−1^	1.82 × 10^−2^	0.55
40	Glyoxylate and dicarboxylatemetabolism	29	8	1.24 × 10^−2^	4.39	2.79 × 10^−1^	1.82 × 10^−2^	0.33
41	Inositol phosphate metabolism	28	5	1.36 × 10^−2^	4.30	2.79 × 10^−1^	1.88 × 10^−2^	0.27
42	Phosphatidylinositol signaling system	26	5	1.37 × 10^−2^	4.29	2.79 × 10^−1^	1.88 × 10^−2^	0.19
43	Citrate cycle (TCA cycle)	20	5	1.37 × 10^−2^	4.29	2.79 × 10^−1^	1.88 × 10^−2^	0.32
44	Fatty acid degradation	37	2	1.41 × 10^−2^	4.26	2.79 × 10^−1^	1.89 × 10^−2^	0.33
45	Thiamine metabolism	22	2	1.52 × 10^−2^	4.19	2.79 × 10^−1^	1.99 × 10^−2^	0
46	Aminoacyl-tRNA biosynthesis	46	13	1.63 × 10^−2^	4.12	2.79 × 10^−1^	2.08 × 10^−2^	0.11
47	Pantothenate and biosynthesis CoA	23	2	1.66 × 10^−2^	4.10	2.79 × 10^−1^	2.08 × 10^−2^	0.15
48	Tyrosine metabolism	16	2	1.90 × 10^−2^	3.96	2.79 × 10^−1^	2.33 × 10^−2^	0.22
49	Amino sugar and nucleotide sugar metabolism	50	7	1.93 × 10^−2^	3.95	2.79 × 10^−1^	2.33 × 10^−2^	0.35
50	Riboflavin metabolism	11	3	1.98 × 10^−2^	3.92	2.79 × 10^−1^	2.34 × 10^−2^	0.07
51	Propanoate metabolism	20	1	2.17 × 10^−2^	3.83	2.79 × 10^−1^	2.46 × 10^−2^	0.06
52	Fatty acid biosynthesis	56	1	2.17 × 10^−2^	3.83	2.79 × 10^−1^	2.46 × 10^−2^	0
53	Glycolysis/Gluconeogenesis	26	4	2.34 × 10^−2^	3.75	2.79 × 10^−1^	2.57 × 10^−2^	0.04
54	Pentose phosphate pathway	19	6	2.35 × 10^−2^	3.75	2.79 × 10^−1^	2.57 × 10^−2^	0.41
55	Fructose and mannose metabolism	20	1	2.42 × 10^−2^	3.72	2.79 × 10^−1^	2.60 × 10^−2^	0.04
56	Lysine biosynthesis	9	3	2.84 × 10^−2^	3.56	2.79 × 10^−1^	2.99 × 10^−2^	0
57	Valine, leucine and isoleucine biosynthesis	22	1	3.59 × 10^−2^	3.33	2.79 × 10^−1^	3.72 × 10^−2^	0
58	Pyruvate metabolism	22	4	3.97 × 10^−2^	3.23	2.79 × 10^−1^	4.04 × 10^−2^	0.31
59	Lysine degradation	18	2	4.05 × 10^−2^	3.21	2.79 × 10^−1^	4.05 × 10^−2^	0

^a^ The pathway analysis was accomplished by Metaboanalyst 4.0 (https://www.metaboanalyst.ca/, the details of the analysis are available in Appendix A), the affected pathways are numbered as in Figure 6; ^b^ annotation of metabolic pathways for identified pea seed metabolites (Figure 6, Appendix A) relied on Kyoto Encyclopedia of Genes and Genomes (KEGG) pathway library of the model plant *Arabidopsis thaliana* L; ^c^ the total number of compounds (Total Cmpd) in the pathway; ^d^ the number of actually matched hits (Hits) from the uploaded identified metabolite data (Appendix A); ^e^ the original *p* value (Raw *p*) calculated from the enrichment analysis, details on the analysis provided in Appendix A; ^f^ the Holm *p* is the *p* value adjusted by Holm-Bonferroni method; ^g^ the *p* value adjusted for False Discovery Rate (FDR *p*); ^h^ the pathway impact value calculated from the pathway topology analysis, details of the analysis are provided in Appendix A.

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
