# Peer review of "Does Protein Glycation Impact on the Drought-Related Changes in Metabolism and Nutritional Properties of Mature Pea (Pisum sativum L.) Seeds?"

_ijms, 2020, doi:10.3390/ijms21020567_

Round 1

Reviewer 1 Report

The present work is devoted to the study of drought-related changes in the metabolism and nutritional properties of mature pea seeds in the context of protein glycation.

The authors carried out a huge experimental work. In general, the work produces a generally positive impression. However, there are several questions to the work:

How drought conditions affect the formation of AGE’s. And if AGE’s are formed, which ones?

What will be the changes in metabolism, if pea seeds are drought for a week?

How will metabolism change if pea seeds are not inoculated with a rhizobial culture?

In the section (Metabolomics analysis of pea seeds) the authors present the differences in metabolites between the control and the test sample. However, the authors should write a conclusion about the nature (positive, negative, neutral) of such changes.

This manuscript can undoubtedly be accepted for publication after revision.

Author Response

We thank the reviewer for the thoughtful review and highly appreciate the valuable comments and suggestions to improve the manuscript. Following these advices we performed all required changes in corresponding sections, as indicated in the following rebuttal addressing each aspect.

Remarks

Remark 1: “How drought conditions affect the formation of AGE’s. And if AGE’s are formed, which ones?

Answer: The requested information is provided. The appropriate changes in text are done:

“Recently, we demonstrated that drought results in glycation at specific arginine and lysine sites in Arabidopsis leaf proteins, yielding mostly Nɛ-(carboxymethyl)lysine (CML), Nδ-(carboxymethyl)arginine and glyoxal-derived hydroimidazolone (Glarg) [ ]” (lines 101 -103).

Remark 2: “What will be the changes in metabolism, if pea seeds are drought for a week?

Answer: This question is still to be answered in the experiments, currently planned in our lab. Based on the knowledge on the metabolism of filling pea seeds under drought conditions, decrease in protein and lipid contents, accompanied with up-regulation of sugar metabolism, can be expected. This might increase seed glycation levels. This is the object of our current work and behind the scope of this manuscript. The changes in text are done:   

“It can be expected that longer times (up to 1 week - 10 days) of drought application might result in more pronounced changes in metabolism of mature seeds. Thus, a pronounced decrease in protein and lipid contents, accompanied with increase in sugar contents, can be expected [ ]. As was shown for Arabidopsis leaf, these changes can be accompanied with increase of glycation levels [ ]. However, for mature seeds this aspect is still to be addressed in future studies” (lines 459 - 464).

Remark 3: “How will metabolism change if pea seeds are not inoculated with a rhizobial culture?

Answer: To the best of our knowledge, these studies were not done so far. However, it is known, that pea plants, inoculated with rhizobia demonstrate higher seed protein accumulation. Thereby, pea lines more responsive to rhizobia demonstrate higher increase in protein contents. Thus, stronger metabolic responses can be expected in this case. The changes in the text are provided

“…which might be dependent on responsivity of pea cultivars to rhizobial symbiont” (lines 454 - 455).

Remark 4: “In the section (Metabolomics analysis of pea seeds) the authors present the differences in metabolites between the control and the test sample. However, the authors should write a conclusion about the nature (positive, negative, neutral) of such changes.

Answer: We assume, that these changes in accumulation of reserve substances indicate adaptive response of plants to drought. The changes in the text are done:

“These changes in accumulation of reserve substances might indicate adaptive response of plants to drought” (lines 453 - 454).

Reviewer 2 Report

This manuscript is about the accumulation of anti-nutritive protein-bound advanced glycation end products (AGEs), triggered by a short-term experimental drought. The points noted by the authors in Discussion seem to be intriguing to readers of IJMS. The reviewer will put a few comments or requests below.

Although the authors focused on the accumulation of AGEs in Introduction, especially in its last paragraph, the manuscript title provides an impression of metabolomic traits instead of particular proteins. The reviewer recommends the revision of the title.

In Discussion, the section 3-1 should be moved into Introduction, because its content does not seem to be based on the results of the present research. The facts or speculations based on results should be discussed.

In the section 3-2, the reviewer feels that the results of the pathway analysis are insufficiently discussed. The authors should associate their points discussed in the section with the distinct names of the pathways.

Author Response

We thank the reviewer for the thoughtful review and highly appreciate the valuable comments and suggestions to improve the manuscript. Following these advices we performed all required changes in corresponding sections, as indicated in the following rebuttal addressing each aspect.

Remarks

Remark 1: “Although the authors focused on the accumulation of AGEs in Introduction, especially in its last paragraph, the manuscript title provides an impression of metabolomic traits instead of particular proteins. The reviewer recommends the revision of the title

Answer: We agree with the reviewer and change the title appropriately:

“Does protein glycation impact on the drought-related changes in metabolism and nutritional properties of mature pea (Pisum sativum L.) seeds?”

Remark 2: “In Discussion, the section 3-1 should be moved into Introduction, because its content does not seem to be based on the results of the present research. The facts or speculations based on results should be discussed.

Answer: The changes are done according to the reviewer’s instructions:

“Indeed, multiple key metabolites of the primary plant metabolic pathways are known as efficient glycation agents or can readily yield pro-glycative compounds during their oxidative degradation [24]. … Addressing protein glycation, one need to keep in mind, however, that plant cells are rich in secondary metabolites - phenolics, terpenes and alkaloids, which need to be considered as potential protein modification agents as well [40–42]. Not less importantly, these secondary metabolites can interfere with glycation and oxidation, directly affecting protein modification rates [43]. Indeed, it is well-known, that the people, consuming exclusively plant-derived foods (vegetarians) are featured with increased AGE blood levels [44]. However, such individuals are less suffering from inflammation-related diseases like atherosclerosis and diabetes mellitus [45]. Thus, on one hand, protein glycation needs to be considered in the context of the seed metabolome dynamics. On the other - one needs to keep in mind that, due to high chemical diversity of plant metabolites, glycation and oxidation represent only one of multiple possible protein modifications in plants.

These considerations brought us to the conclusion, that mild transient drought conditions (which are difficult to avoid in field) can trigger metabolic alterations in seeds, which might either suppress glycation of seed proteins, or/and affect their pro-inflammatory properties in mammals. Despite of the importance of this aspect, it was not appropriately studied so far. Therefore, here we address the question, if the oxidative stress and metabolic alterations, triggered by a short-term experimental drought, can result in accumulation of anti-nutritive protein-bound AGEs, potentially affecting biological properties of mature seeds and potentially triggering pro-inflammatory responses. As in mammals dietary glycation adducts are absorbed in intestine in form of amino acid derivatives or, as was proved for the most representative AGE - CML, dipeptides [46,47], our workflow relied on quantitative solubilization of the total protein fraction followed with its exhaustive enzymatic hydrolysis according to the well-established protocol of Glomb and co-workers [34]. Recently, we demonstrated the compatibility of this workflow with solubilization of proteins and developed an adequate purification technique to make hydrolyzates applicable to cell assays [48]. Thus, our experimental setup can be considered as a simplified model of food digestion in vitro. In this prove of the concept experiment we refused for comprehensive modeling of food digestion conditions, as described by Hellwig et al [47], to preserve the patterns of pea seed AGEs as rich, as possible. Having this methodology in hand, we employed SH-SY5Y human neuroblastoma cells to address the effect of drought on the inflammatory response, triggered by pea seed protein” (lines 99 - 101and 109 – 136).

Remark 3: “In the section 3-2, the reviewer feels that the results of the pathway analysis are insufficiently discussed. The authors should associate their points discussed in the section with the distinct names of the pathways.

Answer: The changes in the section are done in agreement with the reviewer’s suggestions. The discussed points are associated with the pathway names as suggested. The corresponding changes in the text are done:

“Surprisingly, the central metabolic pathways (glycolysis, citrate cycle, pentose phosphate cycle) were distinctly suppressed in seeds (Figure 6 and Table 1)” (lines 431 - 433).

“Indeed, the pathway analysis revealed significant suppression of phenylalanine, tyrosine, tryptophan, histidine, arginine, proline, glycine and serine biosynthesis pathways (Figure 6 and Table 1)” (lines 441 - 443).

“The effect on polysaccharide metabolism can be illustrated by changes in starch, sucrose, fructose, mannose and amino sugar metabolism, as was illustrated by pathway analysis (Figure 6 and Table 1)” (lines 445 - 447).